# Matroid Algorithms Under Size-Sensitive Independence Oracles

**Kiarash Banihashem** [* 1]  **Mohammad Hajiaghayi** [* 1]  **Mahdi JafariRaviz** [* 1]  **Danny Mittal** [* 1]

## Abstract

The standard oracle model for matroid algorithms assumes that each independence query can be answered in constant time, regardless of the size of the queried set. While this abstraction has underpinned much of the theoretical progress in matroid optimization, it masks the true computational effort required by these algorithms. In particular, for natural and widely studied classes such as graphic matroids, even a single independence query can require work linear in the size of the set, making the constant-time assumption implausible. We address this gap by introducing a size-sensitive cost model where the cost of a query $Q$ scales with $|Q|$. Nearly linear-time oracle implementations exist for broad families of matroids, and this refined abstraction therefore captures the true cost of query evaluation while allowing for a more faithful comparison between general matroids and their natural special cases. Within this framework we study three fundamental algorithmic tasks: finding a basis of a matroid, approximating its rank, and approximating its partition size. We establish tight results, proving nearly matching upper and lower bounds that show the optimal query cost is (up to logarithmic factors) quadratic in the size of the matroid. On the algorithmic side, our upper bounds are realized by explicit procedures that construct the desired solution. On the complexity side, our lower bounds are unconditional and already hold even for weaker distinguishing formulations of the problems. Finally, for matroids with maximum circuit size at most $c$, we show that the quadratic barrier can be broken, providing an algorithm that calculates the maximum-weight basis with expected query cost $\mathcal{O}(n^{2-1/c} \log n)$.

## 1. Introduction

Matroid constraints have become a standard modeling tool in the machine learning community for problems that require selecting subsets subject to combinatorial constraints. Rather than encoding a single rigid structure, matroids provide a flexible abstraction that captures a wide range of practically relevant constraints while still admitting algorithmic guarantees. As a result, matroid-based formulations appear across diverse ML settings, including bandit and online learning problems with feasibility constraints (Papadigenopoulos & Caramanis, 2021; Tzeng et al., 2024), various formulations of submodular optimization (Han et al., 2020; Duetting et al., 2022; 2023; Gu et al., 2023), and optimization problems arising in preference elicitation, allocation, and incentive design (Benabbou et al., 2021; Barman & Verma, 2022; Gourvès et al., 2013; Cong et al., 2025).

The study of algorithms for matroids has traditionally been framed in terms of the *number of oracle queries* required to solve a given problem. In this standard model, we have access to an oracle that can determine whether or not any given set is independent in the matroid. Each query to this oracle is treated as having unit cost, i.e., $\mathcal{O}(1)$ time, regardless of the size of the set being queried. This has become the prevailing abstraction in the literature, enabling clean analyses of combinatorial and optimization problems on matroids (Chekuri & Quanrud, 2016; Chakrabarty et al., 2019; Blikstad et al., 2021; 2023; Terao, 2024; Khanna et al., 2025).

However, this abstraction is not always realistic once oracle evaluation itself becomes a computational bottleneck. In practice, answering a feasibility or independence query often requires work proportional to the size of the queried set, since operations that examine large sets cannot be completed in constant time. Yet the oracle model assigns $\mathcal{O}(1)$ cost to all queries regardless of their input size. This creates a mismatch between the abstract query complexity and the actual runtime needed to implement these algorithms, and it means that the true cost of matroid algorithms is not a priori clear for important special cases such as graphs, where an independence query may already require time linear in the size of the queried set. Recent work in the machine learning community has also highlighted the importance of accounting for the computational cost of oracle access

---
*Equal contribution  [1]Department of Computer Science, University of Maryland, College Park, USA. Correspondence to: Mahdi JafariRaviz <mahdij@umd.edu>.

*Proceedings of the 43rd International Conference on Machine Learning*, Seoul, South Korea. PMLR 306, 2026. Copyright 2026 by the author(s).

in matroid optimization, albeit from a different perspective that augments the oracle model (Eberle et al., 2024).

This motivates a more refined complexity model: what happens if we account explicitly for the cost of queries as a function of their size, rather than assuming they are constant time? In this paper we adopt precisely this viewpoint. We initiate a systematic study of *matroid algorithms under size-sensitive query costs*, with a focus on understanding both algorithmic techniques and inherent limitations in this more realistic model. Formally, we consider a model where an oracle can check the independence of a set $S$ in time $|S|$. As we show in this paper, such oracles can be constructed for a broad class of matroids. In this new model, we analyze three fundamental problems:

- Finding a basis of a matroid,

- Approximating the rank of a matroid,

- Computing or approximating the partition size of a matroid.

In all of these cases we obtain *almost-tight upper and lower bounds*. Our upper bounds are achieved by algorithms that explicitly find the output solution (e.g., finding a basis of a matroid or a set of maximum rank), while our lower bounds are unconditional and already hold for the weaker distinguishing versions of the problems. For graphic matroids, these tasks specialize to classical graph problems: bases correspond to spanning forests, and partition size corresponds to graph arboricity.

We distinguish rank estimation from basis finding because the two tasks play complementary roles in our results. Our lower bounds are proved for rank approximation and therefore also apply to the stronger task of basis finding. Conversely, our positive results for basis finding also determine the rank, since the rank is simply the size of any basis returned by the algorithm.

Beyond their intrinsic interest, our results clarify the true cost of matroid algorithms once query evaluation is modeled accurately. This brings the complexity analysis of matroid algorithms closer to practice, and also improves the comparison with important special cases such as graphic matroids. In particular, our upper bounds are stated in terms of the total size of the queried sets. Therefore, for broad families of matroids that admit linear-time oracles, these size-sensitive query-cost bounds translate directly into the same asymptotic running-time bounds.

### 1.1. Our Contributions

We present two main lower bounds for basic matroid tasks in the *size-sensitive query-cost* model. Given a matroid

$M = (E, \mathcal{I})$, in this model the algorithm may submit independence queries $Q \subseteq E$ to an independence oracle; the oracle answers whether $Q \in \mathcal{I}$, and the total cost of the algorithm is the sum of $|Q|$ over all queries made. The theorems below show that, when query cost grows with query size, both rank and matroid-partition-size estimation require quadratic total query cost in the worst case. In Appendix B, we show oracles for *partition*, *graphic*, *bicircular*, *convex transversal*, and *simple job scheduling matroids* which are linear or near-linear in the query size.

#### 1.1.1. BOUNDS FOR RANK ESTIMATION

Given a matroid $M = (E, \mathcal{I})$, $\mathrm{rank}(M)$ is the maximum size of any independent set. A trivial algorithm can find the rank using $\Theta(n^2)$ query cost. The theorem below shows that this bound is tight in general.

**Theorem 1.1.** *Any (possibly randomized) algorithm that, with probability at least $2/3$, approximates the rank of a matroid within a multiplicative factor of $1 \pm 1/40$ must incur $\Omega(n^2)$ query cost in the worst case, where $n$ is the size of the ground set.*

The proof constructs a family of hard instances and shows that any algorithm that uses small total query-size cost cannot distinguish the instances. Fix an integer $m$ and let the ground set be $[3m]$. For every $m$-subset $S \subseteq [3m]$ define $M_{m,S}$ to be the matroid obtained as the union of the free matroid on $S$ and the uniform matroid of rank $m$ on $T = [3m] \setminus S$. The matroid $M_{m,S}$ has rank $2m$ and the key property that every set of size at most $m$ is independent, so queries of size at most $m$ give no information about $S$. For a small constant $\epsilon$ (we take $\epsilon = 1/10$) truncate $M_{m,S}$ to rank $2m - \epsilon m$ to obtain $M'_{m,S,\epsilon}$. Any witness that distinguishes $M_{m,S}$ from $M'_{m,S,\epsilon}$ must be large, and a combinatorial counting argument shows a fixed large set $W$ can be a witness for only a small fraction of choices of $S$. A deterministic algorithm that makes at most $q$ large queries inspects at most $2^{q+1}$ candidate sets, so unless $q = \Omega(m)$ it fails with high probability on a uniformly random $S$. Since each useful query has size $\Theta(m)$ and $n = 3m$, this forces total query cost $\Omega(m^2) = \Omega(n^2)$. Applying Yao's principle extends the deterministic lower bound on this hard distribution to randomized algorithms and yields Theorem 1.1. We refer to Lemma 3.2 for details on Yao's principle.

We also study the problem under the additional assumption that the matroid has bounded circumference (the maximum size of any circuit). In this setting, we design a randomized algorithm with expected query cost strictly below quadratic. Moreover, the algorithm solves a stronger problem: it not only outputs the rank and a basis, but also computes a maximum-weight basis.

**Theorem 1.2.** *For any positive integer $c$, there exists a randomized algorithm (Algorithm 1) that finds the maximum-*

*weight basis of any matroid with circumference at most $c$ while incurring expected total query cost $\mathcal{O}(n^{2-1/c} \log n)$.*

Bounded circumference guarantees that for any element $e$ outside the maximum-weight basis, the fundamental circuit of $e$ has size at most $c$. If a random subset contains this circuit, the algorithm identifies and removes $e$ as the minimum-weight element. Repeating this sampling $O(n \ln n)$ times eliminates non-basis elements with high probability. A final step removes any residual dependent elements.

### 1.1.2. BOUNDS FOR PARTITION SIZE ESTIMATION

Given a matroid $M = (E, \mathcal{I})$, the partition size $k(M)$ is the minimum integer $\alpha$ such that $E$ can be partitioned into $S_1, S_2, \ldots, S_\alpha$ with $S_i \in \mathcal{I}$ for all $i$.

**Theorem 1.3.** *Let $A$ be a (possibly randomized) algorithm that, given a matroid $M$ whose partition size is either $3$ or $4$, determines the partition size of $M$ with probability at least $2/3$. Let $n$ be the number of elements of $M$. Then $A$ incurs $\Omega(n^2)$ query cost in the worst case.*

The proof follows the same scheme as Theorem 1.1 but uses partition matroids together with an $l$-relaxation to ensure that all small sets are automatically independent, forcing any informative query to be large. Given a matroid $M$, its $l$-relaxation $M'$ is defined so that a set $S$ is independent in $M'$ if there exists $S' \subseteq S$ with $|S \setminus S'| \leq l$ and $S' \in \mathcal{I}$. Fix integers $m, \alpha$ and let $n = (\alpha + 1)m$. For a partition $S$ of $[n]$ into $m$ parts of size $\alpha + 1$, let $P_S$ be the partition matroid that allows at most one element from each part. Apply the $l$-relaxation with $l = \frac{m}{\alpha}$ so that every set of size at most $l$ is independent, obtaining $Q_{m,\alpha,S}$, and then truncate its rank by one to get $Q'_{m,\alpha,S}$. The only difference between $Q_{m,\alpha,S}$ and $Q'_{m,\alpha,S}$ occurs on sets of size exactly $m + \frac{m}{\alpha}$, and a set $W$ is a *witness* precisely when it has this size and intersects every part of $S$.

A counting argument shows that a fixed set $W$ can be a witness for only a $\text{poly}(m, \alpha)\gamma(\alpha)^{-m}$ fraction of all partitions, where $\gamma(\alpha) > 1$ for $\alpha \geq 3$. Thus any decision tree that makes at most $q$ large queries explores at most $2^{q+1}$ candidate sets and finds a witness with negligible probability unless $q = \Omega(m)$. Since each useful query has size at least $l = \Theta(n)$, this gives total query cost $\Omega(m^2) = \Omega(n^2)$. Yao's principle then extends the bound from deterministic decision trees to randomized algorithms.

To obtain an algorithm and upper bound for matroid partitioning under the size-sensitive query-cost model, we apply results from Quanrud (2024). They address the base-covering problem: given a matroid $M = (E, \mathcal{I})$ and an integer $k$, compute $k$ bases $S_1, \ldots, S_k$ such that $S_1 \cup \cdots \cup S_k = E$. This relates to the partition problem: $M$ admits a cover by $k$ bases if and only if the partition size is at most $k$.

Quanrud (2024) show that this problem is solvable with $\tilde{\mathcal{O}}(nk)$ independence queries. A direct application of their algorithm results in a query cost of $\tilde{\mathcal{O}}(n^2k)$, as each query can be of size $n$. Instead, we truncate the matroid to rank $\lceil n/k \rceil$ and apply their method. This reduces the total query cost to $\tilde{\mathcal{O}}(n^2)$, which nearly matches the lower bound in Theorem 1.3.

**Theorem 1.4.** *There is an algorithm incurring $\tilde{\mathcal{O}}(n^2)$ query cost to calculate the partition size of a matroid.*

### 1.2. Extension to General Cost Functions

We extend our analysis to a model where the cost of a query $Q$ is $f(|Q|)$ for a non-decreasing function $f$. We show that our lower bounds hold in this setting.

**Theorem 1.5.** *Suppose the cost of querying a set $Q$ is determined by a non-decreasing function $f(|Q|)$. Any (possibly randomized) algorithm that, with probability at least $2/3$, approximates the rank of a matroid within a multiplicative factor of $1 \pm 1/40$ must incur $\Omega(n \cdot f(n/3))$ query cost in the worst case, where $n$ is the size of the ground set.*

**Theorem 1.6.** *Suppose the cost of querying a set $Q$ is determined by a non-decreasing function $f(|Q|)$. Let $A$ be a (possibly randomized) algorithm that, given a matroid $M$ whose partition size is either $3$ or $4$, determines the partition size of $M$ with probability at least $2/3$. Let $n$ be the number of elements of $M$. Then $A$ incurs $\Omega(n \cdot f(n/12))$ query cost in the worst case.*

Notably, if $f(n/12) = \Theta(f(n))$ (e.g., if $f$ is a polynomial), these lower bounds simplify to $\Omega(n \cdot f(n))$.

## 2. Related Work

Matroids are a central abstraction in theoretical computer science, arising in diverse areas such as combinatorial optimization, approximation algorithms, and complexity theory. They have been studied extensively across a variety of settings including online algorithms, dynamic algorithms, and parallel computation (Reif & Spirakis, 1980; Gabow & Tarjan, 1979; Gabow & Westermann, 1988; Kleinberg & Weinberg, 2012; Ehsani et al., 2018; Babaioff et al., 2018; Chakrabarty et al., 2019; Dütting et al., 2023; Banihashem et al., 2024; Buchbinder & Feldman, 2024; Khanna et al., 2025). The study of efficient algorithms for matroids has traditionally focused on oracle-based query models. The standard model is the independence oracle, which allows one to determine in $\mathcal{O}(1)$ time whether a set is independent. A stronger variant is the rank oracle, which instead returns the rank of a queried set (e.g., see (Lee et al., 2015; Chakrabarty et al., 2019)). While these abstractions enable clean analyses, their limitation is that in practice verifying independence requires operations scaling with the size of the set. In contrast, we explicitly account for this issue by

assuming that an independence query on $S$ takes $\Theta(|S|)$ time, and we show that such oracles can be implemented for natural classes of matroids.

A closely related work studies dynamic oracles (Blikstad et al., 2023), where queries can be answered in constant time provided they differ from a previously queried set by the addition or removal of a single element. This requires the oracle to retain state across queries, and can lead to different behavior from the stateless model we study here. For example, the standard greedy algorithm for finding a basis can have much smaller cost in such a dynamic model, while it has quadratic query cost in our model. Thus, our lower-bound proofs are specific to the stateless size-sensitive model and do not directly extend to dynamic oracles. In our view, a memoryless model–where query cost depends only on the size of the set–is also natural and realistic, especially in distributed or parallel systems where different queries may be processed independently or in different machines. Stateless interaction also aligns more closely with lightweight frameworks such as REST APIs.

Finally, many important combinatorial structures can be expressed as special cases of matroids, including graphic and laminar matroids. Corresponding problems in these settings, such as computing the arboricity of a graph (which corresponds to the partition size in the associated matroid) have been widely studied (Gabow & Westermann, 1988; Gabow, 1998; Eden et al., 2020b;a; Ghaffari & Grunau, 2024; de Vos & Christiansen, 2025). Since we show a linear-time oracle can indeed be implemented for numerous matroid classes, algorithms developed in our framework for general matroids immediately yield algorithms for these special cases as well, with the same running time.

## 3. Preliminaries

We use standard set and asymptotic notation. For a positive integer $n$ we let $[n] = \{1, 2, \ldots, n\}$. Multinomial and binomial coefficients are written in the usual way, e.g., $\binom{n}{k}$ and $\binom{n}{k_1, \ldots, k_t}$. We abbreviate polynomial factors by $\mathrm{poly}(\cdot)$; for example $\mathrm{poly}(n)$ denotes an expression polynomial in $n$.

Stirling's approximation will be used in counting arguments; we record the form we use:

$$n! = \Theta\left(\sqrt{n}\left(\frac{n}{e}\right)^n\right) = \mathrm{poly}(n)\left(\frac{n}{e}\right)^n.$$

A *matroid* is a pair $M = (E, \mathcal{I})$ where $E$ is a finite ground set and $\mathcal{I} \subseteq 2^E$ is a family of *independent* sets satisfying: (i) $\emptyset \in \mathcal{I}$, (ii) (downward-closed) if $I \in \mathcal{I}$ and $J \subseteq I$ then $J \in \mathcal{I}$, (iii) (exchange) if $I, J \in \mathcal{I}$ and $|I| < |J|$ then there exists $e \in J \setminus I$ with $I \cup e \in \mathcal{I}$. The *rank* of a matroid $M$, denoted $\mathrm{rank}(M)$, is the maximum size of an independent set in $\mathcal{I}$. A *basis* of $M$ is an independent set of size $\mathrm{rank}(M)$. A *circuit* is a minimal dependent set $C \subseteq E$; that is, $C \notin \mathcal{I}$, but for every element $e \in C$, the set $C \setminus \{e\}$ is independent. For a basis $B$ and an element $e \notin B$, the *fundamental circuit* of $e$ with respect to $B$ is the unique circuit contained in $B \cup \{e\}$. The maximum size of any circuit in $M$ is the *circumference* of $M$. For a set $S \subseteq E$, an element $e$ is in the *span* of $S$ if $e \in S$ or if adding $e$ to $S$ creates a circuit containing $e$.

We recall several standard matroid constructions used in the paper:

- The *uniform matroid* $U_{r,E}$ of rank $r$ on ground set $E$ has independent sets $I \subseteq E : |I| \leq r$.

- The *free matroid* on a subset $S \subseteq E$ is the matroid on ground set $S$ whose family of independent sets is $2^S$ (every subset is independent).

- A *partition matroid* induced by a partition $E = \bigcup_j E_j$ and capacities $c_j$ has independent sets $I$ with $|I \cap E_j| \leq c_j$ for all $j$. In particular, the partition matroid we use has capacity $1$ on each part.

- The *matroid union* of two matroids $M_1 = (E_1, \mathcal{I}_1)$ and $M_2 = (E_2, \mathcal{I}_2)$ is the matroid $M_1 \vee M_2$ whose elements are $E_1 \cup E_2$, and independent sets are all unions $I_1 \cup I_2$ with $I_1 \in \mathcal{I}_1$ and $I_2 \in \mathcal{I}_2$.

- The *truncation* of a matroid $M = (E, \mathcal{I})$ to rank $r$ is the matroid $M^{\leq r} = (E, \mathcal{I}^{\leq r})$ where $\mathcal{I}^{\leq r} = \{I \in \mathcal{I} : |I| \leq r\}$. Truncation reduces the rank to at most $r$ and preserves independence for sets of size $\leq r$.

We formalize the size-sensitive independence oracle used throughout the paper.

**Definition 3.1** (Size-sensitive independence oracle). An algorithm accesses a matroid $M = (E, \mathcal{I})$ via an independence oracle. A query to the oracle is any set $Q \subseteq E$. The oracle returns a Boolean value indicating whether $Q \in \mathcal{I}$. The *size-sensitive cost* of a single query $Q$ is $|Q|$. The total query cost of an algorithm is the sum of the costs of all queries it issues.

To obtain lower bounds against randomized algorithms, we apply Yao's principle (Yao, 1977) in the standard way.

**Lemma 3.2** (Yao's principle, informal). *To prove a lower bound on the expected cost of any randomized algorithm that succeeds with probability at least $p$ on the worst-case instance, it suffices to exhibit a distribution $\mathcal{D}$ over instances such that every deterministic algorithm that succeeds with probability at least $p$ on $\mathcal{D}$ must incur the claimed cost.*

We will also use the notion of a decision tree to model deterministic adaptive algorithms: each internal node corre-

sponds to a queried set and branches according to the oracle answer; leaves correspond to outputs.

# 4. Rank and Basis

In this section we study the problem of finding the rank of a matroid under the linear query cost model. Given a matroid $M = (E, \mathcal{I})$, the goal is to determine the rank of $M$, i.e., the size of a maximum independent set. We note that our lower bounds extend to the problem of finding a basis, as basis construction is at least as hard as rank computation.

A straightforward algorithm maintains an initially empty set $S$ and, for each $e \in E$, queries whether $S \cup e$ is independent; if so, it adds $e$ to $S$. Under the linear query cost model, this procedure incurs a total query cost of $\Theta(n^2)$, identifying a basis and thus the rank. We show that this cost is essentially optimal: any algorithm, including randomized algorithms, must incur $\Omega(n^2)$ query cost in the worst case. Moreover, achieving any constant-factor approximation strictly better than $1 \pm 1/40$ still requires $\Omega(n^2)$ query cost; hence permitting approximation does not reduce the asymptotic query cost.

## 4.1. Lower Bound

We restate Theorem 1.1 below and prove it in the rest of this section.

**Theorem 1.1.** *Any (possibly randomized) algorithm that, with probability at least $2/3$, approximates the rank of a matroid within a multiplicative factor of $1 \pm 1/40$ must incur $\Omega(n^2)$ query cost in the worst case, where $n$ is the size of the ground set.*

To prove the theorem we consider a family of matroids defined as follows.

**Definition 4.1.** Given a positive integer $m$ and a subset $S \subseteq [3m]$ with $|S| = m$, let $T = [3m] \setminus S$. Define $M_{m,S}$ to be the matroid on the ground set $[3m]$ obtained as the matroid union of the free matroid on $S$ and the uniform matroid of rank $m$ on $T$.

**Lemma 4.2.** *Every subset of the ground set of $M_{m,S}$ of size at most $m$ is independent.*

An immediate consequence is that, when $m$ is known but $S$ is unknown, any query of size at most $m$ can never distinguish among different choices of $S$; such queries reveal no new information about $S$. Hence only queries of size greater than $m$ are potentially informative. Since $n = 3m$, any non-redundant query has size at least $m = \Theta(n)$. We will next consider truncations of $M_{m,S}$.

**Definition 4.3.** Let $m, S$ be as above and let $\epsilon \in (0, 1)$ be such that $\epsilon m$ is an integer. Define $M'_{m,S,\epsilon}$ to be the truncation of $M_{m,S}$ to rank $2m - \epsilon m$. That is, the independent

sets of $M'_{m,S,\epsilon}$ are exactly those independent sets of $M_{m,S}$ whose size is at most $2m - \epsilon m$.

Note that $M_{m,S}$ has rank $2m$, so $M'_{m,S,\epsilon}$ reduces the rank by $\epsilon m$. To formalize the difficulty of distinguishing $M_{m,S}$ from its truncation, we introduce the notion of a *witness* set.

**Definition 4.4.** Given $m$, $\epsilon$, and $S$ as above, a set $W \subseteq [3m]$ is a *witness* if $W$ is independent in $M_{m,S}$ but not independent in $M'_{m,S,\epsilon}$.

**Lemma 4.5.** *A set $W$ is a witness for $S$ if and only if $|W| > 2m - \epsilon m$ and $|W \setminus S| \leq m$.*

Thus any witness must be larger than $2m - \epsilon m$ but may contain at most $m$ elements outside $S$.

We next bound, for a fixed $W$, how many choices of $S$ admit $W$ as a witness.

**Lemma 4.6.** *Fix $m, \epsilon$ and let $W \subseteq [3m]$ be a witness. If $|W| = 2m - \delta m$ for some $\delta \geq 0$, then $\delta < \epsilon$, and the number of sets $S$ of size $m$ for which $W$ is a witness is at most*

$$\binom{2m - \delta m}{m - \delta m} \binom{2m + \delta m}{\delta m}.$$

*In particular, for every $\delta \in [0, \epsilon)$ this quantity is at most $\binom{2m}{m} \binom{2m + \epsilon m}{\epsilon m}$.*

Define a distribution $\mathcal{D}_{m,\epsilon}$ over instances as follows: choose $S$ uniformly at random from all $m$-subsets of $[3m]$, and then with probability $1/2$ take the instance to be $M_{m,S}$ and with probability $1/2$ take it to be $M'_{m,S,\epsilon}$.

The following lemma quantifies the inability of deterministic algorithms that make few useful queries to succeed on $\mathcal{D}_{m,\epsilon}$.

**Lemma 4.7.** *Let $A$ be any deterministic algorithm that, when run on an instance drawn from $\mathcal{D}_{m,\epsilon}$, makes at most $q$ queries of size greater than $m$ (all other queries may be ignored as they yield no information). If $A$ attempts to approximate the rank to within a factor of $\epsilon/4$, then $A$ succeeds with probability at most*

$$\frac{1}{2} + \frac{2^q \cdot \binom{2m}{m} \binom{2m + \epsilon m}{2m}}{\binom{3m}{m}}.$$

We now convert the bound above into a lower bound on the number of useful queries.

**Lemma 4.8.** *Fix $\epsilon \leq 1/10$. Any deterministic algorithm $A$ that, on instances drawn from $\mathcal{D}_{m,\epsilon}$, outputs a rank estimate that is within a factor of $\epsilon/4$ with probability at least $2/3$ must make $\Omega(m)$ queries of size more than $m$.*

Finally, apply Yao's principle to obtain Theorem 1.1.

*Proof of Theorem 1.1.* Suppose for contradiction that there exists a randomized algorithm that, with probability at least $2/3$, approximates the rank within factor $1/40$ while paying $o(n^2)$ query cost in the worst case. By Yao's principle there must then exist a deterministic algorithm that achieves success probability at least $2/3$ on the distribution $\mathcal{D}_{m,\epsilon}$ while paying $o(n^2)$ worst-case query cost, for suitable choice of parameters. Take $\epsilon = 1/10$ and $n = 3m$. By Lemma 4.8, any deterministic algorithm that succeeds on $\mathcal{D}_{m,\epsilon}$ with probability at least $2/3$ must make $\Omega(m)$ queries of size more than $m$, and therefore pay query cost $\Omega(m^2) = \Omega(n^2)$. This contradicts the assumption that the randomized algorithm pays $o(n^2)$ query cost; hence no such randomized algorithm exists. $\qquad\square$

### 4.2. Better Algorithms for Bounded Circumference

In this section, we consider the problem of finding a maximum-weight basis. Given a matroid $M = (E, \mathcal{I})$ with weights $w_e$ for each $e \in E$, the goal is to identify a basis of $M$ that maximizes the total weight.

The main obstacle that causes quadratic query cost in the size-sensitive query model is the need to make large queries to determine dependence. For example, an element $e$ not in the maximum-weight basis may only be spanned by a large set of other elements. Formally, this means that for some independent set $I$ not containing $e$, the fundamental circuit of $e$ with respect to $I$ is large. This motivates the study of whether faster algorithms are possible when all circuits are small, that is, when the circumference is bounded.

*Circumference* is defined as the maximum size of a circuit in the matroid. We present an algorithm to compute a maximum-weight basis when this parameter is bounded. This result contrasts with Theorem 1.1, which demonstrates that estimating the rank in the general case requires quadratic query cost.

Without loss of generality, we assume all weights are pairwise distinct. We can enforce this by ordering the elements $e_1, \ldots, e_n$ and mapping each weight $w_{e_i}$ to $W \cdot w_{e_i} + i$, where $W$ is sufficiently large. This transformation ensures the maximum-weight basis is unique.

We provide Algorithm 1. The algorithm begins with the full set of elements and iteratively removes those not in the optimal basis. In each step, it samples a random subset $S$. If $S$ is dependent, the algorithm sorts the items by decreasing weight and uses binary search to identify the shortest prefix that is dependent. The last element of this prefix is then removed. We repeat this random sampling to discard most incorrect items. Finally, we apply this same removal procedure to the remaining set until we obtain the final basis.

In the rest of this section we prove Theorem 1.2, restated

---

**Algorithm 1** Algorithm for maximum-weight basis in matroids with circumference at most $c$.

**Input:** Matroid $M$ with circumference at most $c$, set of elements $E = \{e_1, \ldots, e_n\}$, and distinct weights $w_e$ for each $e \in E$.

**Output:** The maximum-weight basis.

Initialize $B \leftarrow E$.

**for** $t = 1, \ldots, n \ln n$ **do**

    Construct $S \subseteq B$ by inserting each $e \in B$ independently with probability $n^{-1/c}$.

    **while** $S$ is not independent **do**

        Order $S$ by decreasing weight as $s_1, \ldots, s_{|S|}$.

        Binary search to find the least $j$ such that $\{s_1, \ldots, s_j\}$ is dependent.

        Remove $s_j$ from both $S$ and $B$.

    **end while**

**end for**

**while** $B$ is not independent **do**

    Order $B$ by decreasing weight as $b_1, \ldots, b_{|B|}$.

    Binary search to find the least $j$ such that $\{b_1, \ldots, b_j\}$ is dependent.

    Remove $b_j$ from $B$.

**end while**

**return** $B$.

---

below.

**Theorem 1.2.** *For any positive integer $c$, there exists a randomized algorithm (Algorithm 1) that finds the maximum-weight basis of any matroid with circumference at most $c$ while incurring expected total query cost $\mathcal{O}(n^{2-1/c} \log n)$.*

*Proof of Theorem 1.2.* Let $B^*$ denote the maximum-weight basis and let $D = E \setminus B^*$ be the set of elements not in $B^*$. For each $d \in D$ fix a fundamental circuit $C_d$ of $d$ with respect to $B^*$; by the circumference bound we have $|C_d| \leq c$. Moreover, every element of $C_d \setminus \{d\}$ has weight larger than the weight of $d$.

Fix $d \in D$. In any iteration of the outer for-loop, each element of $B$ is inserted into $S$ independently with probability $n^{-1/c}$, so the probability that all elements of $C_d$ appear in $S$ is at least

$$\Pr\left[C_d \subseteq S\right] = \left(n^{-1/c}\right)^{|C_d|} \geq n^{-1}.$$

If $C_d \subseteq S$, then when $S$ is ordered by decreasing weight, the element $d$ is the least-weight element of the dependent prefix $s_1, \ldots, s_j$ found by the binary search, and hence $d$ will be removed from $B$ during that iteration. Therefore, the probability that $d$ survives all $n \ln n$ independent iterations of the for-loop is at most

$$\left(1 - n^{-1}\right)^{n \ln n} \leq e^{-\ln n} = \frac{1}{n}.$$

By linearity of expectation, the expected number of elements of $D$ that remain in $B$ after the for-loop is 1.

We next bound the expected query cost. Consider queries performed during the for-loop. For a fixed iteration, the random set $S$ has expected size $|S| = |B| \cdot n^{-1/c} \leq n^{1-1/c}$, so each independence query on a subset of $S$ has expected cost $n^{1-1/c}$. The for-loop makes one initial independence query per iteration to test $S$, and each time the inner while-loop finds a dependent prefix it performs $O(\log |S|) \leq O(\log n)$ independence queries for the binary search and one extra query to re-evaluate the while condition. Each removal performed by the inner loop corresponds to removing a distinct element from $B$, and thus across the entire execution the inner loop removes at most $n$ elements. Hence the total number of independence queries issued during the for-loop is $O(n \log n)$, each with expected size at most $n^{1-1/c}$. Therefore, the expected total query cost contributed by the for-loop is $O(n^{2-1/c} \log n)$.

Finally, consider the final while-loop. By the expectation bound above, the number of elements of $B$ that are outside $B^*$ after the for-loop is 1 in expectation. Each iteration of the final while-loop removes one element from $B$ and performs $O(\log n)$ independence queries on prefixes of $B$, each of cost at most $n$. Hence, the expected query cost of the final loop is $O(n \log n)$, which is dominated by the cost from the for-loop.

Combining these bounds yields the claimed expected total query cost $O(n^{2-1/c} \log n)$. This completes the proof.

We note that the correctness follows because an element $d$ is not in $B^*$ if and only if it is lightest member of a circuit. Throughout the algorithm, we only discard elements with this property, and thus the final basis is exactly $B^*$. $\square$

## 5. Matroid Partitioning

In this section we consider the matroid partitioning problem: given a matroid $M = (S, \mathcal{I})$, find the minimum integer $\alpha$ such that $S$ can be partitioned into subsets $S_1, \ldots, S_\alpha$ with each $S_i$ independent.

### 5.1. Lower Bound

We prove Theorem 1.3, restated below.

**Theorem 1.3.** *Let $A$ be a (possibly randomized) algorithm that, given a matroid $M$ whose partition size is either 3 or 4, determines the partition size of $M$ with probability at least $2/3$. Let $n$ be the number of elements of $M$. Then $A$ incurs $\Omega(n^2)$ query cost in the worst case.*

Hence computing the partition size requires quadratic query cost; in particular, even a $(1+\epsilon)$-approximation for any constant $\epsilon < 1/3$ requires quadratic query cost. The struc-

ture of the proof is similar to the lower bound for estimating the rank in Theorem 1.1; the main difference is the matroid family used to prove the bound. We begin with a simple partition matroid.

**Definition 5.1.** Given nonnegative integers $n, m$ and a partition of $[n]$ into parts $S = (S_1, \ldots, S_m)$, let $P_S$ be the partition matroid in which a set $I$ is independent iff it contains at most one element from each $S_j$.

We will consider positive integers $m, \alpha$ with $n = (\alpha + 1)m$ and where all parts $S_j$ have equal size $\alpha + 1$. Such a $P_S$ can be partitioned into $\alpha + 1$ independent parts. The next step is a transformation that makes all sufficiently small sets independent.

**Lemma 5.2.** *Let $M = (E, \mathcal{I})$ be a matroid and let $l$ be an integer. Define $M' = (E, \mathcal{I}')$ by declaring $S \subseteq E$ to be in $\mathcal{I}'$ iff there exists $L \subseteq S$ with $|L| \leq l$ and $S \setminus L \in \mathcal{I}$. Then $M'$ is a matroid.*

This transformation ensures that every set of size at most $l$ in $M'$ is independent, so queries of size at most $l$ provide no information about the structure beyond independence of small sets. We now define the family of matroids we use.

**Definition 5.3.** Fix positive integers $m, \alpha$ with $\alpha$ dividing $m$, and let $S$ be a partition of $[(\alpha + 1)m]$ into $m$ parts of equal size. Let $Q_{m,\alpha,S}$ be the matroid obtained by applying Lemma 5.2 to the partition matroid $P_S$ with $l = \frac{m}{\alpha}$.

Note that $Q_{m,\alpha,S}$ has partition size $\alpha$: within each part of $S$, one can place the first $\alpha$ elements into distinct parts of the independent partition, and the remaining $m$ elements can be distributed evenly, adding $\frac{m}{\alpha}$ elements to each part. Since each part is a basis, this shows that the partition size is exactly $\alpha$.

We then apply a truncation similar to that in Definition 4.3.

**Definition 5.4.** Fix positive integers $m, \alpha$ with $\alpha$ dividing $m$, and let $S$ be a partition of $[(\alpha + 1)m]$ into $m$ equal parts. Let $Q'_{m,\alpha,S}$ be the truncation of $Q_{m,\alpha,S}$ to rank $m + \frac{m}{\alpha} - 1$.

Since $Q_{m,\alpha,S}$ has rank $m + \frac{m}{\alpha}$, the truncation $Q'_{m,\alpha,S}$ reduces the rank by one. The partition of $Q_{m,\alpha,S}$ into $\alpha$ independent sets was perfect (each part was a basis), so $Q'_{m,\alpha,S}$ is not $\alpha$-partitionable since its bases were truncated. Therefore, any algorithm that distinguishes $\alpha$-partitionable matroids from $(\alpha + 1)$-partitionable ones must distinguish $Q_{m,\alpha,S}$ from $Q'_{m,\alpha,S}$. We now define witnesses for this difference.

**Definition 5.5.** For $W \subseteq [(\alpha + 1)m]$, say $W$ is a *witness* to a partition $S$ of $[(\alpha + 1)m]$ into $m$ equal parts if $W$ is independent in $Q_{m,\alpha,S}$ but not in $Q'_{m,\alpha,S}$.

**Lemma 5.6.** *$W$ is a witness for $S$ iff $|W| = m + \frac{m}{\alpha}$ and $W$ contains at least one element from each part of $S$.*

A single witness is useful for only a small fraction of partitions, as quantified below.

**Lemma 5.7.** *For given $m, \alpha$, a set $W \subseteq [(\alpha+1)m]$ is a witness for at most*

$$m! \binom{m + \frac{m}{\alpha}}{m} \binom{\alpha m}{\alpha, \ldots, \alpha}$$

*distinct partitions $S$.*

The total number of partitions of $[(\alpha+1)m]$ into $m$ equal parts is $\binom{(\alpha+1)m}{\alpha+1, \ldots, \alpha+1}$. Combining this with the previous lemma gives the following lemma. For convenience, we define the following function:

$$\gamma(\alpha) = \left(1 + \frac{1}{\alpha}\right)^{\alpha - 1 - \frac{1}{\alpha}} \left(\frac{1}{\alpha}\right)^{\frac{1}{\alpha}}. \tag{1}$$

**Lemma 5.8.** *For given $m, \alpha$, any set $W \subseteq [(\alpha+1)m]$ is a witness for at most a*

$$\gamma(\alpha)^{-m} \operatorname{poly}(m, \alpha)$$

*fraction of all partitions $S$.*

For $\alpha \geq 3$, $\gamma(\alpha) = \left(1 + \frac{1}{\alpha}\right)^{\alpha - 1 - \frac{1}{\alpha}} \left(\frac{1}{\alpha}\right)^{\frac{1}{\alpha}}$ satisfies $\gamma(\alpha) > 1.1199 > 1$, so the fraction covered by a single witness is exponentially small in $m$.

We now apply Yao's principle. Define a distribution $\mathcal{D}_{\alpha,m}$ over instances by choosing a partition $S$ of $[(\alpha+1)m]$ into $m$ equal parts uniformly at random, and then choosing $Q_{m,\alpha,S}$ or $Q'_{m,\alpha,S}$ each with probability $1/2$. The following lemma shows that an algorithm must find a witness to distinguish these two matroids.

**Lemma 5.9.** *Let $A$ be any deterministic algorithm that makes at most $q$ queries of size greater than $\frac{m}{\alpha}$. When the input is sampled from $\mathcal{D}_{\alpha,m}$, $A$ outputs the correct answer with probability at most*

$$\frac{1}{2} + \frac{2^q \operatorname{poly}(m, \alpha)}{\gamma(\alpha)^m}.$$

Having the above lemma, we are now ready to prove Theorem 1.3.

*Proof of Theorem 1.3.* Set $\alpha = 3$. By Lemma 5.9, any deterministic algorithm that succeeds with probability at least $\frac{2}{3}$ on $\mathcal{D}_{\alpha,m}$ must make $q$ useful queries satisfying

$$\frac{1}{2} + \frac{2^q \operatorname{poly}(m, \alpha)}{\gamma(\alpha)^m} \geq \frac{2}{3},$$

which implies $2^q \geq \frac{1}{6} \operatorname{poly}(m, \alpha) \gamma(\alpha)^m$ and hence $q \geq m \lg(\gamma(\alpha)) - o(m)$. Since $\gamma(3) > 1$, we have $q = \Theta(m)$.

Each useful query has size at least $\frac{m}{\alpha}$, so the worst-case query cost is at least $\Omega(m \cdot \frac{m}{\alpha}) = \Omega(m^2)$.

By Yao's principle, the same lower bound applies to randomized algorithms: any randomized algorithm that pays $o(m^2)$ query cost would yield, via its best deterministic strategy against $\mathcal{D}_{\alpha,m}$, a deterministic algorithm that succeeds with probability at least $\frac{2}{3}$ while making $o(m)$ useful queries, contradicting the bound above. Translating $m$ back to $n = (\alpha+1)m$ yields the stated $\Omega(n^2)$ worst-case query cost. □

### 5.2. Upper Bound

We will prove Theorem 1.4, restated below.

**Theorem 1.4.** *There is an algorithm incurring $\tilde{\mathcal{O}}(n^2)$ query cost to calculate the partition size of a matroid.*

To prove the above, we will use the following result from Quanrud (2024).

**Lemma 5.10** (Quanrud (2024), Theorem 4.4). *For integer capacities, one can compute either a covering of $(1 + \epsilon)\lambda$ bases or a certificate of infeasibility for any covering of $(1 - \epsilon)\lambda$ bases in time bounded by $\tilde{\mathcal{O}}(n/\epsilon)$ independence queries.*

The next simple truncation fact shows that we never need to query sets larger than $\lceil n/k \rceil$ when $k$ is the partition size.

**Lemma 5.11.** *Let $M = (E, \mathcal{I})$ be a matroid on $n$ elements that can be partitioned into $k$ independent sets. Let $M'$ be the truncation of $M$ to rank $r = \lceil n/k \rceil$. Then $M'$ can be partitioned into $k$ independent sets as well.*

The above lemmas are all the tools needed to prove Theorem 1.4. We refer to Appendix A for the full proof.

## 6. Extension to General Cost Functions

Consider a general model where querying a set of size $k$ costs $f(k)$ for a non-decreasing function $f$. The standard greedy algorithm computes a basis using $n$ queries. Since each query has size at most $n$, the total query cost is $O(n \cdot f(n))$.

We show that this greedy algorithm is close to optimal. Recall the hard instances from Section 4, where every set of size $m = n/3$ is independent. Consequently, any query $Q$ with $|Q| \leq m$ provides no information. To distinguish between instances, an algorithm must query sets larger than $m$.

For the rank problem, Lemma 4.8 implies that any successful algorithm must perform $\Omega(n)$ queries of size strictly greater than $m$. Thus, the total query cost is lower bounded by $\Omega(n \cdot f(n/3))$.

Similarly, for the partition size, Lemma 5.9 requires $\Omega(n)$

queries of size at least $m/\alpha$. Since $\alpha \leq 4$, the total query cost is lower bounded by $\Omega(n \cdot f(n/12))$.

The proofs for Theorems 1.5 and 1.6 follow the proofs of Theorems 1.1 and 1.3 closely, with the above mentioned difference in the cost of each useful query.

## 7. Conclusion

We studied matroid algorithms in a model where the cost of an independence query is the size of the queried set. For rank estimation and partition size, we proved nearly tight quadratic upper and lower bounds. We also showed that this barrier can be broken for matroids with bounded circumference.

These results show that accounting for query size can change how the cost of matroid algorithms should be understood, even for basic tasks. Several directions remain open, including identifying other settings where faster algorithms are possible and understanding how size-sensitive query costs affect broader matroid optimization problems.

## Acknowledgements

The work is partially supported by DARPA expMath, ONR MURI 2024 award on Algorithms, Learning, and Game Theory, Army-Research Laboratory (ARL) Grant W911NF2410052, NSF AF:Small grants 2218678, 2114269, 2347322.

## Impact Statement

This paper presents work whose goal is to advance the field of Machine Learning. There are many potential societal consequences of our work, none of which we feel must be specifically highlighted here.

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

## A. Omitted Proofs

We provide proofs omitted from the main body in this section. We also restate the theorem/lemma statement first.

**Lemma 4.6.** *Fix $m, \epsilon$ and let $W \subseteq [3m]$ be a witness. If $|W| = 2m - \delta m$ for some $\delta \geq 0$, then $\delta < \epsilon$, and the number of sets $S$ of size $m$ for which $W$ is a witness is at most*

$$\binom{2m - \delta m}{m - \delta m}\binom{2m + \delta m}{\delta m}.$$

*In particular, for every $\delta \in [0, \epsilon)$ this quantity is at most $\binom{2m}{m}\binom{2m+\epsilon m}{\epsilon m}$.*

*Proof of Lemma 4.6.* If $W$ is a witness, then by Lemma 4.5 $|W| > 2m - \epsilon m$, so $\delta < \epsilon$. Further, $W$ must contain at most $m$ elements outside $S$, which implies that $S$ contains at least $m - \delta m$ elements from $W$.

To upper bound the number of possible $S$, choose exactly $m - \delta m$ elements of $S$ from $W$ (in $\binom{2m - \delta m}{m - \delta m}$ ways) and call it $X$. We then choose the remaining $\delta m$ elements of $S$ from $[3m] \setminus X$, which has size $3m - (m - \delta m) = 2m + \delta m$ (in $\binom{2m + \delta m}{\delta m}$ ways). Note that this will double count some sets $S$, but that is okay since we are providing an upper bound. This yields the stated bound. Finally, each binomial factor is monotone in the appropriate argument, so replacing $\delta$ by $\epsilon$ gives the stated uniform upper bound. $\qquad \square$

**Lemma 4.7.** *Let $A$ be any deterministic algorithm that, when run on an instance drawn from $\mathcal{D}_{m,\epsilon}$, makes at most $q$ queries of size greater than $m$ (all other queries may be ignored as they yield no information). If $A$ attempts to approximate the rank to within a factor of $\epsilon/4$, then $A$ succeeds with probability at most*

$$\frac{1}{2} + \frac{2^q \cdot \binom{2m}{m}\binom{2m+\epsilon m}{2m}}{\binom{3m}{m}}.$$

*Proof of Lemma 4.7.* The decision process of $A$ can be viewed as a binary tree of depth at most $q$, which has at most $2^{q+1}$ nodes and therefore at most $2^{q+1}$ distinct sets $W$ that $A$ ever queries. By Lemma 4.6, each such $W$ is a witness for at most $\binom{2m}{m}\binom{2m+\epsilon m}{2m}$ different choices of $S$. Hence the total number of $S$ for which $A$ queries some witness is at most $2^{q+1}\binom{2m}{m}\binom{2m+\epsilon m}{2m}$, so the probability (over the random choice of $S$) that $A$ ever queries a witness is at most $2^{q+1} \cdot \binom{2m}{m}\binom{2m+\epsilon m}{2m}/\binom{3m}{m}$.

Condition on the event that $A$ does not query any witness for the chosen $S$. In that event the answers to all queries that $A$ sees are identical whether the instance is $M_{m,S}$ or $M'_{m,S,\epsilon}$, so $A$ must output the same rank estimate in both cases. Since the ranks of these two matroids are $2m$ and $2m - \epsilon m$ respectively, any single estimate can be correct for at most one of the two possibilities, since $\frac{2m - \epsilon m}{2m} = 1 - \epsilon/2 < 1 - \epsilon/4$. Thus conditioned on this event $A$ has success probability at most $1/2$. Combining the two cases yields that $A$'s overall success probability is at most

$$\frac{2^{q+1}\binom{2m}{m}\binom{2m+\epsilon m}{2m}}{\binom{3m}{m}} + \left(1 - \frac{2^{q+1}\binom{2m}{m}\binom{2m+\epsilon m}{2m}}{\binom{3m}{m}}\right) \cdot \frac{1}{2} = \frac{1}{2} + \frac{2^q\binom{2m}{m}\binom{2m+\epsilon m}{2m}}{\binom{3m}{m}},$$

as claimed. $\qquad \square$

**Lemma 4.8.** *Fix $\epsilon \leq 1/10$. Any deterministic algorithm $A$ that, on instances drawn from $\mathcal{D}_{m,\epsilon}$, outputs a rank estimate that is within a factor of $\epsilon/4$ with probability at least $2/3$ must make $\Omega(m)$ queries of size more than $m$.*

*Proof of Lemma 4.8.* By Lemma 4.7 and the requirement that the success probability is at least $2/3$, the number $q$ of useful queries must satisfy

$$\frac{1}{2} + \frac{2^q\binom{2m}{m}\binom{2m+\epsilon m}{2m}}{\binom{3m}{m}} \geq \frac{2}{3},$$

so

$$2^q \geq \frac{1}{6} \cdot \frac{\binom{3m}{m}}{\binom{2m}{m}\binom{2m+\epsilon m}{2m}}.$$

We now analyze the term $T = \frac{\binom{3m}{m}}{\binom{2m}{m}\binom{2m+\epsilon m}{2m}}$ asymptotically. We first algebraically manipulate it as follows:

$$T = \frac{\binom{3m}{m}}{\binom{2m}{m}\binom{2m+\epsilon m}{2m}} = \frac{\frac{(3m)!}{m!(2m)!}}{\frac{(2m)!}{m!m!} \cdot \frac{(2m+\epsilon m)!}{(2m)!(\epsilon m)!}} = \frac{(3m)!m!(\epsilon m)!}{(2m)!((2+\epsilon)m)!}.$$

We then note that by Stirling's approximation, we have that $n! = \Theta(\sqrt{n}(\frac{n}{e})^n)$. Applying this to the above expression gives:

$$T = \Theta\left(\frac{\sqrt{3m}(\frac{3m}{e})^{3m}\sqrt{m}(\frac{m}{e})^m\sqrt{\epsilon m}(\frac{\epsilon m}{e})^{\epsilon m}}{\sqrt{2m}(\frac{2m}{e})^{2m}\sqrt{(2+\epsilon)m}(\frac{(2+\epsilon)m}{e})^{(2+\epsilon)m}}\right)$$

$$= \Theta\left(\sqrt{m}\frac{(\frac{3m}{e})^{3m}(\frac{m}{e})^m(\frac{\epsilon m}{e})^{\epsilon m}}{(\frac{2m}{e})^{2m}(\frac{(2+\epsilon)m}{e})^{(2+\epsilon)m}}\right) = \Theta\left(\sqrt{m}\left(\frac{3^3\epsilon^\epsilon}{2^2(2+\epsilon)^{2+\epsilon}}\right)^m\right).$$

We therefore must have $2^q \geq \frac{1}{6}T = \Theta\left(\sqrt{m}\left(\frac{3^3\epsilon^\epsilon}{2^2(2+\epsilon)^{2+\epsilon}}\right)^m\right)$. When $\epsilon \leq \frac{1}{10}$, we have that $\frac{3^3\epsilon^\epsilon}{2^2(2+\epsilon)^{2+\epsilon}} \geq 1.128$, meaning that we must have $2^q \geq \Theta(\sqrt{m}\cdot 1.128^m)$. It therefore follows that we must have $q \geq \lg\sqrt{m}+m\lg(1.128)-O(1) = \Omega(m)$ as desired. $\qquad\square$

**Lemma 5.2.** *Let $M = (E, \mathcal{I})$ be a matroid and let $l$ be an integer. Define $M' = (E, \mathcal{I}')$ by declaring $S \subseteq E$ to be in $\mathcal{I}'$ iff there exists $L \subseteq S$ with $|L| \leq l$ and $S \setminus L \in \mathcal{I}$. Then $M'$ is a matroid.*

*Proof of Lemma 5.2.* We verify the matroid exchange property for $\mathcal{I}'$. Let $S, T \in \mathcal{I}'$ with $|S| < |T|$. By definition there exist subsets $L \subseteq S, R \subseteq T$ with $|L| \leq l, |R| \leq l$, and $S \setminus L, T \setminus R \in \mathcal{I}$. Choose $L$ to be minimal with this property.

If $|L| < l$, pick any $t \in T \setminus S$. Then $L \cup t$ has size at most $l$, and hence $(S \cup t) \setminus (L \cup t) = S \setminus L \in \mathcal{I}$, so $S \cup t \in \mathcal{I}'$.

If $|L| = l$, then $|S \setminus L| = |S| - l < |T| - l \leq |T \setminus R|$. Since $M$ is a matroid, there exists $t \in (T \setminus R) \setminus (S \setminus L)$ with $(S \setminus L) \cup t \in \mathcal{I}$. Note $t \notin L$ because otherwise $L \setminus t$ would be a smaller set contradicting minimality of $L$. Thus $t \notin S$, and using the same $L$ we obtain $S \cup t \in \mathcal{I}'$. $\qquad\square$

**Lemma 5.6.** *$W$ is a witness for $S$ iff $|W| = m + \frac{m}{\alpha}$ and $W$ contains at least one element from each part of $S$.*

*Proof of Lemma 5.6.* Since $Q'_{m,\alpha,S}$ is the truncation of $Q_{m,\alpha,S}$ from rank $m + \frac{m}{\alpha}$ to $m + \frac{m}{\alpha} - 1$, a set can be independent in one and not the other only if it has size $m + \frac{m}{\alpha}$. For such a set $W$, $W$ is independent in $Q_{m,\alpha,S}$ iff there exists $L \subseteq W$ with $|L| \leq \frac{m}{\alpha}$ and $W \setminus L \in P_S$. The latter means $W \setminus L$ contains at most one element from each part of $S$, so $|W \setminus L| \leq m$. Given $|W| = m + \frac{m}{\alpha}$, both inequalities must be tight, so $|L| = \frac{m}{\alpha}$ and $|W \setminus L| = m$. Hence $W \setminus L$ has exactly one element from each part of $S$, which implies $W$ contains at least one element from every part. The converse is immediate. $\qquad\square$

**Lemma 5.7.** *For given $m, \alpha$, a set $W \subseteq [(\alpha + 1)m]$ is a witness for at most*

$$m!\binom{m + \frac{m}{\alpha}}{m}\binom{\alpha m}{\alpha, \ldots, \alpha}$$

*distinct partitions $S$.*

*Proof of Lemma 5.7.* By Lemma 5.6, if $W$ is a witness for $S$ then $W$ contains at least one element from each part of $S$. To form such an $S$ we first choose $m$ distinct elements of $W$ and assign them to the $m$ parts (there are $m!\binom{m+\frac{m}{\alpha}}{m}$ ways to do this), and then partition the remaining $\alpha m$ elements of $[(\alpha + 1)m]$ into $m$ parts of size $\alpha$ (there are $\binom{\alpha m}{\alpha, \ldots, \alpha}$ ways). This yields the stated upper bound; some partitions may be counted multiple times. $\qquad\square$

**Lemma 5.8.** *For given $m, \alpha$, any set $W \subseteq [(\alpha+1)m]$ is a witness for at most a*

$$\gamma(\alpha)^{-m} \operatorname{poly}(m, \alpha)$$

*fraction of all partitions $S$.*

*Proof of Lemma 5.8.* It suffices to estimate the ratio

$$\frac{\binom{(\alpha+1)m}{\alpha+1, \ldots, \alpha+1}}{m! \binom{m+\frac{m}{\alpha}}{m} \binom{\alpha m}{\alpha, \ldots, \alpha}} = \gamma(\alpha)^m \operatorname{poly}(m, \alpha)^{-1}.$$

Expanding the multinomial coefficients and canceling factorials gives

$$\frac{\binom{(\alpha+1)m}{\alpha+1, \ldots, \alpha+1}}{m! \binom{m+\frac{m}{\alpha}}{m} \binom{\alpha m}{\alpha, \ldots, \alpha}} = \frac{((\alpha+1)m)! \left(\frac{m}{\alpha}\right)!}{(\alpha+1)^m \left(m+\frac{m}{\alpha}\right)! (\alpha m)!}.$$

Applying Stirling's approximation $n! = \operatorname{poly}(n)(n/e)^n$ yields

$$\frac{((\alpha+1)m)! \frac{m}{\alpha}!}{(\alpha+1)^m (m+\frac{m}{\alpha})! (\alpha m)!} = \operatorname{poly}(m, \alpha) \cdot \frac{\left(\frac{(\alpha+1)m}{e}\right)^{(\alpha+1)m} \left(\frac{\frac{m}{\alpha}}{e}\right)^{\frac{m}{\alpha}}}{(\alpha+1)^m \left(\frac{m+\frac{m}{\alpha}}{e}\right)^{m+\frac{m}{\alpha}} \left(\frac{\alpha m}{e}\right)^{\alpha m}}$$

$$= \operatorname{poly}(m, \alpha) \cdot \left( \frac{(\alpha+1)^{\alpha+1} \left(\frac{1}{\alpha}\right)^{\frac{1}{\alpha}}}{(\alpha+1)\left(1+\frac{1}{\alpha}\right)^{1+\frac{1}{\alpha}} \alpha^{\alpha}} \right)^m$$

$$= \operatorname{poly}(m, \alpha) \cdot \left( \left(1+\frac{1}{\alpha}\right)^{\alpha-1-\frac{1}{\alpha}} \cdot \left(\frac{1}{\alpha}\right)^{\frac{1}{\alpha}} \right)^m,$$

which proves the claim. $\qquad\square$

**Lemma 5.9.** *Let $A$ be any deterministic algorithm that makes at most $q$ queries of size greater than $\frac{m}{\alpha}$. When the input is sampled from $\mathcal{D}_{\alpha,m}$, $A$ outputs the correct answer with probability at most*

$$\frac{1}{2} + \frac{2^q \operatorname{poly}(m, \alpha)}{\gamma(\alpha)^m}.$$

*Proof of Lemma 5.9.* Only queries of size more than $\frac{m}{\alpha}$ can reveal a witness; we therefore ignore smaller queries and assume $A$ makes at most $q$ useful queries. The adaptive behavior of $A$ can be viewed as a binary decision tree of depth at most $q$, whose internal nodes correspond to useful queries. Such a tree has fewer than $2^{q+1}$ nodes, so $A$ can query at most $2^{q+1}$ distinct sets $W$. By Lemma 5.8, each queried set is a witness for at most a $\operatorname{poly}(m, \alpha)\gamma(\alpha)^{-m}$ fraction of partitions $S$, so the probability that $A$ ever queries a witness is at most $\frac{2^{q+1} \operatorname{poly}(m,\alpha)}{\gamma(\alpha)^m}$.

Conditioned on selecting a partition $S$ for which $A$ does not query any witness, $A$ cannot distinguish $Q_{m,\alpha,S}$ from $Q'_{m,\alpha,S}$ and thus must output the same answer for both, succeeding with probability $1/2$ over the final coin flip that chooses which matroid was presented. Hence the overall success probability is at most

$$\frac{1}{2}\left(1 - \frac{2^{q+1} \operatorname{poly}(m,\alpha)}{\gamma(\alpha)^m}\right) + 1 \cdot \frac{2^{q+1} \operatorname{poly}(m,\alpha)}{\gamma(\alpha)^m} = \frac{1}{2} + \frac{2^q \operatorname{poly}(m,\alpha)}{\gamma(\alpha)^m},$$

as claimed. $\qquad\square$

**Lemma 5.11.** *Let $M = (E, \mathcal{I})$ be a matroid on $n$ elements that can be partitioned into $k$ independent sets. Let $M'$ be the truncation of $M$ to rank $r = \lceil n/k \rceil$. Then $M'$ can be partitioned into $k$ independent sets as well.*

*Proof of Lemma 5.11.* Let $S_1, \ldots, S_k$ be an independent partition of $E$ in $M$. If $|S_i| \leq r$ for every $i$ then the same partition is independent in $M'$, and we are done. Otherwise there exist indices $i, j$ with $|S_i| > r$ and $|S_j| < r$. Then by the matroid exchange property, there must exist an element $e \in S_i \setminus S_j$ such that $S_j \cup e$ is independent. We will move this $e$ into $S_j$. By repeating this process, we will arrive at partition $S'_1, \ldots, S'_k$ in which every part has size at most $r$.

Since each $S'_i$ has size $\leq r$, truncating $M$ to rank $r$ does not change the independence status of any $S'_i$. Hence $S'_1, \ldots, S'_k$ is an independent partition of $M'$, proving the lemma. $\square$

**Theorem 1.4.** *There is an algorithm incurring $\tilde{\mathcal{O}}(n^2)$ query cost to calculate the partition size of a matroid.*

*Proof of Theorem 1.4.* We determine the partition size $k$ by testing candidate values $\lambda$ via a binary search over the interval $[1, n]$. For each candidate $\lambda$ we perform the following test:

1. Truncate $M$ to rank $r = \lceil n/\lambda \rceil$. By Lemma 5.11, this preserves the property that $M$ can be partitioned into $\lambda$ sets (if true).

2. Run the algorithm from Lemma 5.10 on the truncated matroid with parameter $\lambda$ and $\epsilon = 1/(2\lambda)$ as follows: any query of size at most $r$ is recursively computed via a call to the oracle for $M$, while any query of larger size is immediately answered in the negative. The lemma guarantees one of two outcomes in $\tilde{\mathcal{O}}(n/\epsilon) = \tilde{\mathcal{O}}(n\lambda)$ independence queries:
    - either a covering of at most $\lfloor (1+\epsilon)\lambda \rfloor = \lambda$ bases, showing $k \leq \lambda$,
    - or a certificate of infeasibility for any covering of at most $\lfloor (1-\epsilon)\lambda \rfloor = \lambda - 1$ bases, showing $k \geq \lambda$.

The queries that are passed through to the oracle for $M$ have size at most $r = O(n/\lambda)$. Hence the total size-sensitive cost of one test is

$$\tilde{\mathcal{O}}(n\lambda) \cdot O\left(\frac{n}{\lambda}\right) = \tilde{\mathcal{O}}(n^2).$$

Since binary search performs $O(\log n)$ such tests, the total query cost remains $\tilde{\mathcal{O}}(n^2)$. This establishes the theorem. $\square$

# B. Linear Oracles

In this section we describe linear or near-linear independence oracles for several common matroid families. These oracles are standard; we include them for completeness. The selection of matroids is similar to that in (Quanrud, 2024).

**Partition matroids.** As defined in Section 3, let $M = (E, \mathcal{I})$ be a partition matroid defined by a partition $E = \bigcup_j E_j$ and capacities $c_j$. Given a query $I \subseteq E$, an independence check can be implemented in $\mathcal{O}(|I|)$ time by scanning the elements of $I$, maintaining a counter $d_j$ for the part index $j$ of each element, and returning "dependent" as soon as some $d_j > c_j$.

**Graphic matroids.** For a graph $G = (V, E)$ the graphic matroid has ground set $E$ and $I \subseteq E$ is independent iff the subgraph $(V, I)$ is a forest. A linear-time oracle (in the size of the query) proceeds by building the induced subgraph on the edges in $I$ and running a graph traversal algorithm on that subgraph. The query is dependent if some component contains at least as many edges as vertices (equivalently, when a cycle is discovered).

**Bicircular matroids.** The bicircular matroid of a graph $G = (V, E)$ has $I \subseteq E$ independent iff each connected component of the induced subgraph $(V, I)$ contains at most one cycle. Using the same induced-subgraph traversal as above, the query is dependent if some component contains strictly more edges than vertices.

**Convex-transversal matroids and simple job-scheduling matroids.** Let $G = (L, R, E)$ be a bipartite graph where $R$ is given a linear order and each $v \in L$ has a neighbourhood that is an interval in $R$. The convex-transversal matroid (Edmonds & Fulkerson, 1965) on ground set $L$ declares $I \subseteq L$ independent iff there exists a matching that covers $I$. A simple job-scheduling matroid is the special case where each neighborhood is a prefix of $R$.

Given a query $I \subseteq L$, represent each $x \in I$ by its interval $[a_x, b_x] \subseteq R$. A correct greedy test is obtained by sorting the intervals by increasing right endpoint $b_x$ (and breaking ties by decreasing left endpoint $a_x$), then assigning each interval

to the smallest unused position $p \in [a_x, b_x]$. Maintaining the set of unused positions can be done with a balanced binary search tree, where consecutive unused elements are collapsed into a single interval. Thus the independence oracle runs in near-linear time in the query size (plus a logarithmic factor for the sorting and the binary search tree).

