# OpenReview forum: "Matroid Algorithms Under Size-Sensitive Independence Oracles"
_ICML.cc/2026/Conference — ICML 2026 spotlight_

### Official Review · Reviewer_ALNj · 2026-02-25

**Soundness:** 4
**Presentation:** 4
**Significance:** 3
**Originality:** 3
**Overall Recommendation:** 5
**Confidence:** 3

**Summary:**

The paper examines two classical tasks on matroids (determining a base/the rank and matroid partitioning) in an augmented model: While matroid algorithms typically measure complexity in terms of the number of independence queries, the authors here assume that each query incurs cost linear in the size of the queried set. This is motivated by a range of linear time query implementations for typical matroid types.

For both the base and partitioning problem, the authors demonstrate that there are algorithms with worst-case cost $O(n^2)$ and $\tilde{O}(n^2)$, respectively, where $n$ is the size of the ground set.
They contrast this by providing lower bounds for the problems (even for randomized approximation algorithms with at least a certain probability of success) of $\Omega(n^2)$ cost.
For finding a maximum (weighted) base, they provide an algorithm showing that the $\Omega(n^2)$ barrier can be broken for instances with smaller circumference.

Last, they generalize their lower bounds to arbitrary  cost functions $f$ for the oracle (beyond linear cost) by establishing lower bounds of $\Omega(n\cdot f(n/3))$ and $\Omega(n\cdot f(n/12))$ for the base and partitioning problem, respectively.

**Compliance With Llm Reviewing Policy:**

Affirmed.

**Final Justification:**

The paper provides sound theoretical answers to a question in the field of matroids by analyzing an extended where queries do not have constant cost. The paper is very well written and mathematically sound. This sound contribution to a relevant field was already enough for me to (weakly) support acceptance. My main concern was the relevance of the chosen model which does not take into account previous queries. However, as it arguably is a straight-forward and natural model to take the size of the query into account and that in both the paper and the rebuttal the author(s) point to (some) cases where being stateless seems a reasonable assumption, I fully support acceptance.

**Key Questions For Authors:**

1. How do you expect your lower bounds to behave under dynamic models, for example when assuming that queries can be answered in constant time if they only slightly differ from a previous query as studied by Blikdstad et al (2023)?

2.  Line 90: what do you mean by "preserving the same time bounds?" Cannot the runtime guarantees increase by up to a factor of $|S|$ with linear-time oracle implementations?

**Limitations:**

yes

**Strengths And Weaknesses:**

The authors examine an interesting question by investigating the actual underlying complexity of algorithms in the matroid query model. They provide a natural and simple expansion of the model capturing some of this complexity and provide a comprehensive picture of the complexity of two natural matroid tasks under this model. The presentation and structure of the results is very clear and easy to follow, the proofs are rigorous and stated in a manner that makes them easy to understand and verify.

Having not much experience in the study of matroids, it is hard for me to judge the impact of the result. To my limited understanding of the field, the results of the paper seem novel and address a natural open question.
The authors do a good job at addressing concerns that might be raised on the limitation of the model to linear time queries, namely by demonstrating that many types of matroids admit such implementations (in the appendix) and generalizing their lower bounds to arbitrary query cost functions (Section 6).


On the other hand, I feel that the paper would benefit from discussing the choice of the model even more. The authors cite Blikstad et al. (2023), who consider a model where query time might depend on previously posed queries. While the authors give good examples on where a stateless model is more plausible, I am curious about which dynamics seem reasonable and whether the general concept of the provided lower bounds might transfer to dynamic settings (see also Question 1)

I checked all proofs - they are conducted carefully and several of them involve some non-trivial constructions and insights. Multiple of them manage to achieve the desired results without experiencing too much technical depth.

-- Detailed Feedback (only minor issues) --

The theorems inconsistently use the terms "cost" and "query cost"

ln 343ff (right) argues that the partition size is at most $\alpha$, not that is precisely $\alpha$. The latter is only argued after Definition 5.4

the operator "poly" is typeset differently when it is in a lemma statement

Formatting of references: some urls go into the right margin

ln 789: The proof might benefit from recalling that all bases have size at most $n/\lambda$, thereby implying that any covering using $\lambda$ independent sets is necessarily a partition.

---

> ### Author Rebuttal · Authors · 2026-03-30
>
> We thank the reviewer for the careful and positive review.
>
> We also thank the reviewer for the detailed minor comments. We will fix these issues in the final version.
>
> Regarding the model of Blikstad et al. (2023), we agree that this comparison deserves more discussion, and we will expand it in the final version. For your question, our current lower-bound proofs are specific to the stateless model studied in this paper and do not extend to the dynamic setting of Blikstad et al. (2023). In that model, queries that differ only slightly from previous queries may be answered much faster. For example, the standard greedy algorithm for finding a basis can be implemented with a cost of $O(n)$ there, whereas its cost is $O(n^2)$ in our model. This difference is one reason we view the two models as capturing different settings. We still emphasize that our model is simpler and more direct, since it does not require maintaining a dynamic oracle state across queries, which can be more complex to implement in practice.
>
> Regarding Question 2, we agree that the wording at line 90 should be clearer. By “preserving the same time bounds,” we mean the following: our upper bounds are already stated in terms of the total size of the queried sets. Therefore, if a matroid class admits an independence oracle that answers a query $Q$ in time $O(|Q|)$, then our size-sensitive cost bounds translate directly into the same running-time bounds. We will revise the sentence to make this precise.
>
> If our clarifications have successfully addressed your main concerns, we would greatly appreciate it if you might consider updating your score.

---

> > ### Author Rebuttal · Reviewer_ALNj · 2026-04-02
> >
> > I thank the author(s) for their response and addressing my questions. I support acceptance at ICML

---

### Official Review · Reviewer_uHTv · 2026-03-12

**Soundness:** 4
**Presentation:** 3
**Significance:** 4
**Originality:** 4
**Overall Recommendation:** 5
**Confidence:** 3

**Summary:**

Most matroid algorithms assume independence can be answered in constant time, but this may not be the case. Under the assumption, that an oracle can check the independence of set $S$ in time $|S|$, the paper studies the task of finding a basis of a matroid, approximating the rank of a matroid, and computing or approximating the partition size of a matroid. The paper proves matching upper and lower bounds and shows that optimal query cost is quadratic up to logarithmic factors in the size of the matroid. Given matroids with matching circuit size $c$, the paper presents an algorithm which does better than quadratic in query cost.

**Compliance With Llm Reviewing Policy:**

Affirmed.

**Final Justification:**

My opinion for this paper has stayed fairly positive throughout the whole review process. This is overall a very technically sound paper with impressive proofs and wide applicability. For submission to a conference like ICML, I do think that the paper could be adjusted to give more context on matroids and their applications, but the paper remains clear even for someone with less background on the topic.

**Key Questions For Authors:**

1. You mentioned graph matroids as a specific example of when an independence query for set $S$ can take $|S|$. Are there notable applications for graph matroids in particular? (Establishing applications in the paper would emphasize the significance of the results in this paper)
2. Are there any other properties which might result in an algorithm with smaller total query cost beyond fixing the circumference to at most $d$? (The circumference property feels particularly special. This question might be obvious to someone with more familiarity with matroids than I, but it does seem interesting and significant if there did or didn't happen to be other properties.)

**Limitations:**

Yes

**Strengths And Weaknesses:**

Soundness: Proofs are given for each claim, making them very sound.

Presentation: The submission could benefit from a conclusion section. In the introduction, it states and lists three main tasks with bullet points which are studied, but throughout the rest of the paper discussing finding the basis and the rank are always essentially grouped together. It might be helpful to group them in the introduction as well. It does feel a bit awkward to state all the main results with matroids in section 1 and not have matroids be explained until section 3.

Significance: The results have broad applications to any situation where matroids are involved and an independence query isn't trivial, which includes graphic matroids. Matroids are relevant to problems in theoretical computer science, combinatorics, and graph theory.

Originality: This paper works in the case where an oracle can check the independence of set $S$ in time $|S|$ instead of $O(1)$ which is a relevant and original assumption that provides a deeper understanding of matroid algorithms.

---

> ### Author Rebuttal · Authors · 2026-03-30
>
> We thank the reviewer for the positive review and careful reading.
>
> Regarding the conclusion section, we agree with the reviewer. We will add a section concluding our results and discussing future directions. Regarding the separation of rank and basis, we do so to emphasize our results in both directions: our negative results are for rank approximation, which naturally hold for basis finding too, and our positive results are for basis finding, which naturally provide the rank. We will make this more explicit in the camera-ready version of the paper.
>
> Below we answer the reviewer’s question:
> > You mentioned graph matroids as a specific example of when an independence query for set $S$ can take $|S|$. Are there notable applications for graph matroids in particular? (Establishing applications in the paper would emphasize the significance of the results in this paper)
>
> Graphic matroids are a main example because an independence query is checking whether a set of edges forms a forest. In this case, finding a basis is finding a spanning forest, and partition size is known as arboricity. We will add this connection more clearly in the introduction to better highlight the significance.
>
> > Are there any other properties which might result in an algorithm with smaller total query cost beyond fixing the circumference to at most $d$? (The circumference property feels particularly special. This question might be obvious to someone with more familiarity with matroids than I, but it does seem interesting and significant if there did or didn't happen to be other properties.)
>
> We thank the reviewer for their great question. Bounded circumference is one clean condition that gives subquadratic cost because dependence can be witnessed by small circuits. We agree that it would be very interesting to identify other structural properties that also lead to faster algorithms. At present, this remains an open direction, and we see it as a natural topic for future work.

---

> > ### Author Rebuttal · Reviewer_uHTv · 2026-04-04
> >
> > I thank reviewers for the thorough rebuttals. My questions are addressed and I maintain my original rating.

---

### Official Review · Reviewer_7VKR · 2026-03-13

**Soundness:** 3
**Presentation:** 4
**Significance:** 2
**Originality:** 2
**Overall Recommendation:** 4
**Confidence:** 4

**Summary:**

The paper aims to understand matroid algorithms with independence oracle where the oracle’s query complexity is size aware. In classical setting, independence oracle is assumed to run in O(1) time, while this paper assumes that independence oracle requires O(|Q|) time where Q is the queried set of elements of the given matroid. Under this size-sensitive cost model, the authors study two problems: rank estimation (finding the size of the maximum feasible set) and partition size estimation (finding the minimum number of feasible sets needed to cover all elements). For rank estimation, they show that classical algorithm is asymptotically optimal by demonstrating \Omega(n^2) lower bound in the size-sensitive cost model. When the size of the largest circuit(circumference) is bounded by c, they provide improved algorithms with total runtime: O(n^{2-1/c} \log n). For the partition size estimation problem, they also demonstrate a lower bound of $\Omega(n^2)$ for the runtime.

The lower bound results utilizes Yao’s minimax principle by constructing a hard probability distribution over matroids. For rank estimation, the construction takes a uniform matroid on 2m elements, adds m free elements so that all feasible sets have size 2m. Then, randomly truncates the rank to either 2m or (2−\epsilon)m. Now, one needs to distinguish whether the matroid has 2m or (2-\epsilon)m. This requires O(m) many samples at least as we need to distinguish whether (2-\epsilon)m rank is caused due to truncation or hitting uniform matroid more than free elements. The partition size estimation follows a similar logic.

**Compliance With Llm Reviewing Policy:**

Affirmed.

**Key Questions For Authors:**

- This is not a question or comment, but the paper feels better suited for a TCS venue such as SODA, ICALP, ESA, or ISAAC, where the target audience is more aligned with the technical content. I found this venue selection unusual.
- Can you elaborate more on whether the algorithms provided for partition size estimation and small circumference cases are novel or not? Whether some variants have already appeared in the literature or not? If some variants of these algorithms are known before, giving better credit would be important.

**Strengths And Weaknesses:**

Strengths:
- Paper is well-written. Notations, use of notation, theorems and proof sketches are well-written.
- It opens a direction to investigate matroid algorithms in a more practical way.
- The theory behind proof are complex enough to even recognized as a publication in TCS conferences indicating that the contributions are fairly novel and non-trivial.

Weaknesses:
- The paperacknowledges prior work on iterative querying models where swapping a single element between consecutive queries reduces the cost to O(1). This seems a very natural model and already being studied. The paper motivates their model by studying parallel computations or using memory-less api access. Matroid optimization with accessing local in-device structure is more common than parallel or API-used scenarios. The motivation for the size-sensitive model feels somewhat artificial as a result.
- Even though the motivations is to introduce a new model for designing algorithms for a more realistic scenario (size-sensitive cost model), the optimal algorithm for rank estimation is the one used in the standard independence oracle model. This paper concludes that existing algorithms are somehow already optimal in the size-sensitive cost model too.
- Building on the previous two points, an artificially motivated model and a conclusion that reduces to known algorithms make it harder to see the relevance of this work to the ML community. This raises questions about venue fit.

---

> ### Author Rebuttal · Authors · 2026-03-30
>
> We thank the reviewer for the careful reading and helpful suggestions.
>
> We agree that dynamic oracle models are also natural; our goal is not to replace them, but to study a different natural setting in which the oracle is stateless. Our contribution is that in this model, the quadratic cost achieved by simple algorithms is in fact unavoidable for general matroids.
>
> We also appreciate the concern about venue fit. Our view is that the paper is relevant to the ICML audience because it is motivated by the practical use of matroid algorithms. For many natural matroid classes, no constant-time independence oracle is known, and this creates a gap between the usual oracle model and the actual cost of using these algorithms in practice. Our size-sensitive model is intended to study this gap directly. Although several of our results are lower bounds, we believe they are still useful in practice because they clarify when the standard oracle abstraction hides a real computational barrier.
>
> On the algorithmic novelty:
>
> - The algorithm for general partition size estimation is not claimed as a new algorithmic primitive. It is an adaptation of Quanrud (2024), as we state in the paper just before Theorem 1.4. Our contribution there is to show that, after truncating the matroid rank, Quanrud's method gives an $\\tilde{O}(n^2)$ bound in the size-sensitive model.
> - For the bounded-circumference result, to the best of our knowledge, the sampling-and-pruning algorithm for finding a maximum-weight basis is new.
>
> If our clarifications have resolved your primary concerns, we would be grateful if you might consider adjusting your score accordingly.

---

> > ### Author Rebuttal · Reviewer_7VKR · 2026-04-01
> >
> > I thank authors for their response. My concerns are addressed. I still think the natural venue for this paper is a theory conference rather than ICML, but I weakly support acceptance since it is well written and has a nice conceptual and theoretical contribution. If other reviewers or the AC think it fits ICML, my weak support stands.

---

### Official Review · Reviewer_fdF4 · 2026-03-19

**Soundness:** 4
**Presentation:** 4
**Significance:** 3
**Originality:** 3
**Overall Recommendation:** 5
**Confidence:** 3

**Summary:**

The standard oracle model for matroid algorithms assumes that each independence query (query about whether a set is independent) can be answered in constant time, regardless of the size of the queried set. This is an unrealistic assumption, and motivated by this this paper introduces an oracle model where the cost of querying a set Q scales with |Q|, rather than assuming constant time. They go on to establish tight upper and lower bounds for fundamental matroid problems such as rank estimation and partition size.

**Compliance With Llm Reviewing Policy:**

Affirmed.

**Final Justification:**

The other reviews and rebuttal reinforced my prior assessment, so I increased my score.

**Key Questions For Authors:**

- What would be the implications of these results on algorithms such as the greedy algorithm for submodular maximization with a matroid constraint? Does it change what is most efficient if we look beyond constant time independence queries?

**Limitations:**

yes

**Strengths And Weaknesses:**

Strengths
- I think the paper is well motivated and appears novel. It is true that in optimization involving matroid constraints queries about the matroid are treated as constant time, but in reality they are not.
- Very clear and well-written.
- Paper has upper and lower bounds on time complexity results.

---

> ### Author Rebuttal · Authors · 2026-03-30
>
> We thank the reviewer for the careful reading and positive assessment. Below we answer their question.
>
> > What would be the implications of these results on algorithms such as the greedy algorithm for submodular maximization with a matroid constraint? Does it change what is most efficient if we look beyond constant time independence queries?
>
> We thank the reviewer for this question. We had not considered submodular maximization under a matroid constraint, but we agree that this is an important direction. In fact, this problem can be viewed as one of the motivations for our model: in submodular optimization, function evaluation is also typically not constant time, so both value queries and independence queries carry real computational cost.
>
> We believe a natural direction for future work is to revisit greedy and related algorithms for submodular optimization in a model that accounts more realistically for both of these costs.
>
> If our response has resolved your main concerns, we kindly ask that you consider updating your score.

---

> > ### Author Rebuttal · Reviewer_fdF4 · 2026-04-04
> >
> > I agree that that sounds like an interesting application of this work. After reading the other reviews and rebuttals, I decided to update my score to accept.

---

### Decision · Program_Chairs · 2026-04-30

**Decision:**

Accept (spotlight)

**Comment:**

This paper addresses an important gap in the existing literature on matroid algorithms in the oracle model, which is the very unrealistic assumption that each oracle call takes constant time, regardless of the size of the set given as the argument. Instead, the paper proposes a new model, where the cost of each oracle call depends on the set size. It provides several upper and lower bounds for fundamental matroid problems.

All the reviewers agree that the paper is well motivated and opens a new direction of research. They also mention that the paper is well written. One of the reviewers comments that the theory introduced in the paper is on a nontrivial technical level, even if it was to be judged by a usually higher standard of theory conferences. This is a strong accept.